# Fermented Maillard Reaction Products by *Lactobacillus gasseri* 4M13 Alters the Intestinal Microbiota and Improves Dysfunction in Type 2 Diabetic Mice with Colitis

**DOI:** 10.3390/ph14040299

**Published:** 2021-03-28

**Authors:** Yu-Jin Jeong, Ho-Young Park, Han-Kyul Nam, Kwang-Won Lee

**Affiliations:** 1Department of Biotechnology, College of Life Sciences and Biotechnology, Korea University, 1, 5-ga, Anam-dong, Sungbuk-gu, Seoul 02841, Korea; ujinny1227@korea.ac.kr (Y.-J.J.); hankyele@naver.com (H.-K.N.); 2Research Division of Food Functionality, Korea Food Research Institute, Wanju 55365, Korea; hypark@kfri.re.kr

**Keywords:** inflammatory bowel disease, type 2 diabetes mellitus, gut microbial composition, fermented Maillard reaction products, biological therapy

## Abstract

Inflammatory bowel disease is a chronic relapsing disease. Multiple factors can cause inflammatory bowel disease (IBD), including diet, imbalance of the immune system, and impaired intestinal barrier function. Type 2 diabetes mellitus is a complex and chronic metabolic disease caused by a combination of insulin resistance and an ineffective insulin secretory response. The co-occurrence of these two diseases, demonstrating interrelated effects within the gut microbiota, has been frequently reported. This study evaluated the effects of a fermented glycated conjugate of whey protein and galactose with *Lactobacillus gasseri* 4M13 (FMRP) to prevent type 2 diabetes mellitus with inflammatory bowel disease. C57BLKS/J- db/db mice were orally administered FMRP for 14 consecutive days and 2% dextran sulfate sodium (DSS) in water ad libitum for 5 days to induce colitis. FMRP-fed mice showed improved insulin secretion and symptoms of colitis. Compared to the DSS group, the FMRP group showed a decreased abundance of six bacterial genera and increased abundance of *Alistipes* and *Hungateiclostridium*. In cecal contents, the levels of short-chain fatty acids increased in the FMRP group compared to those in the DSS group. Continuous administration of FMRP thus may improve the homeostasis of not only insulin secretion and inflammation, but also the intestinal environment in inflammatory bowel disease and type 2 diabetes mellitus.

## 1. Introduction

Inflammatory bowel disease (IBD) consists of Crohn’s disease (CD), which affects the entire intestine, and ulcerative colitis (UC), which only invades the intestinal epithelium [1]. IBD is a chronic recurrent disease caused by inflammation, metabolic stress, and the collapse of the intestinal barrier [2]. IBD is a result of the imbalance of the intestinal immune system and gut microbial dysbiosis attributed to environmental and genetic variables [3]. However, the role of dysbiosis and the mechanisms of host–microbiome relations in IBD are unclear [4]. The prevalence of IBD is constantly increasing worldwide, and it affects healthcare and medication costs [5]. Moreover, IBD debilitates patients physically and psychosocially during their lifetime [5].

Diabetes mellitus (DM), which is typically divided into type 1 DM and type 2 DM (T2DM), is a chronic, relapsing metabolic disorder [6]. T2DM is mainly attributed to factors such as diet, age, lifestyle, and stress, rather than genetic factors [6,7]. Younger IBD patients with microbial disorders are more susceptible to DM than non-IBD patients due to chronic inflammation and increased insulin levels [8]. Increased inflammation and gut microbial dysbiosis lead to both DM and IBD [9,10]. For both disorders, biotherapy research needs to develop incremental treatments with reduced side effects [3,11].

Gut microbiota composition is affected by diet, medicine, host genetics, and diseases [12]. In a previous study, compared to the healthy group, the proportion of microbes belonging to all species comprising the gut microbiota of CD and UC patients was unbalanced and that of *Firmicutes* was enriched [13]. Moreover, the ratio of *Bacteroidetes/Firmicutes* and diversity of gut microbiota have been suggested as the markers for T2DM patients [14]. Altering gut microbial composition is a new approach to improve IBD and DM [3,10,12,13,14,15]. Recently, several attempts have been made to improve IBD and DM by modifying the microbial composition using functional foods and nutritional supplements, such as sugars, fiber, proteins, and probiotics [16].

The Maillard reaction products (MRPs) obtained by a non-enzymatic browning reaction after conjugation of sugars and milk protein exert biological effects such as anti-cardiovascular, anti-hepatotoxic, and anti-inflammatory effects [17,18,19,20]. However, it is also worth noting that the Maillard reaction can result in the formation of advanced glycation end products (AGEs), such as methylglyoxal-derived AGEs, which have been associated to oxidative stress and diabetes [21]. Thus, it is necessary to minimize the quantity of AGEs. Treatment with *Lactobacillus gasseri* strains resulted in weight reduction, demonstrated anti-inflammatory properties, and protected the gut epithelium against dextran sulfate sodium (DSS)-induced rat colitis [22,23]. *Lactobacillus gasseri* 4M13 (L.4M13) strains, which were isolated from infant feces and were used in present study as probiotics, have functional properties such as inhibitory effects on α-glucosidase activity and nitric oxide (NO) production, anti-oxidants and immune enhanced effects [24]. Microbial fermentation of MRPs with probiotics has more powerful effects such as anti-inflammation, antioxidant effects anti-IBD, and anti-microbial activity [25,26,27]. The conjugates demonstrate high potential as prebiotics, and their fermented products obtained by the action of probiotics are more effective [28]. Based on this, we hypothesized that the treatment of fermented MRPs with L.4M13 strains changes gut microbial diversity and composition and improves biological dysfunction in DM and IBD. Although probiotic and fermented dietary supplement treatments play an important role in maintaining the intestinal health of IBD and DM patients, the treatment of IBD and DM in conjunction has not been well studied. Therefore, in this study, we investigated the effects of the fermented glycated conjugate of whey protein and galactose with *Lactobacillus gasseri* 4M13 (FMRP) in T2DM with colitis.

## 2. Results

### 2.1. FMRP (FL, Low Dose of FMRP; FH, High Dose of FMRP) Alters Disease-Related Biomarkers in the Serum of DSS-Induced Colitis in DB/DB Mice

All groups were treated with phosphate buffered saline (PBS) or FMRP orally for 14 d, and the changes in body weight were examined daily after the ad libitum administration of DSS water (Figure 1A). To demonstrate the changes in blood glucose and insulin concentrations in C57BLKS/J- db/db mice (DB/DB) with DSS-induced colitis, the intraperitoneal glucose tolerance test (IPGTT) was performed, and the insulin levels were measured using an ELISA kit (Figure 1B–D). The serum insulin concentration was significantly higher (*p* < 0.05) in the DSS group than the DB group (Figure 1B). However, the insulin concentrations in the FL and FH groups were similar to that of the DB group. The IPGTT curve is shown in Figure 1C, and its area under the curve (AUC) is presented in Figure 1D. A significant difference (*p* < 0.05) was observed only in the FL group compared to the other groups with respect to the IPGTT AUC.

To evaluate the immune pathogenesis of DM with IBD, nitric oxide (NO) and TNF-α levels were measured in the serum (Figure 2). Compared to the DB group, levels of both NO and TNF-α were significantly decreased in the DSS group. There was no significant difference in NO levels between the DSS and FMRP groups (FL and FH) (Figure 2A). In contrast, TNF-α levels in the FMRP groups were significantly (*p* < 0.05) upregulated compared to that in the DSS group (Figure 2B).

### 2.2. FMRP Ameliorates the Symptoms of DSS-Induced Colitis in DB/DB Mice

DSS treatment resulted in reduced bowel length and increased diarrhea and stools [29]. In the wild-type mice with IBD, the administration of FMRP alleviated colonic damage in DSS-induced colitis (submitted for publication). In the present study, to demonstrate the preventive effect of FMRP, colon length shortening, disease activity index (DAI) score, permeability, and myeloperoxidase (MPO) activity were analyzed. The colon length of the DSS group was significantly (*p* < 0.001) lower than that of the DB group, whereas the length of the FH group significantly (*p* < 0.01) increased compared with that of the DSS group (Figure 3A). The DAI score among the groups (DSS/FL/FH) treated with DSS with the treatment initiated 5 d before euthanizing, when the FMRP was administered for 2 weeks, was not significantly different (Figure 3B). To confirm the degree of permeability in the colon, the level of fluorescein isothiocyanate (FITC)-dextran was measured in the blood. The level of FITC-dextran in the DB and FH groups was significantly (*p* < 0.05) lower than that in the DSS group (Figure 3C). The level of MPO activity, representing immune cell infiltration and damage of colon tissue [29], was significantly (*p* < 0.05) higher in the DSS group than that in the DB group. The MPO levels in the FL (*p* = 0.264) and FH groups (*p* = 0.094) were lower than those in the DSS group, although no significant difference was observed (Figure 3D). To evaluate the influence of FMRP on inflammatory factors, the mRNA levels of TNF-α, IL-1β, IL-10, IFN-γ, and iNOS were analyzed using quantitative reverse transcription PCR (qPCR) (Figure 3E). Except for TNF-α, the mRNA levels of other inflammatory factors were significantly (*p* < 0.05) lower in the DSS, FL, and FH groups than those in the DB group. Compared with the DSS group, both the FL and FH groups showed significant (*p* < 0.05) decreases in the mRNA expression of IL-1β, IFN-γ, and iNOS (Figure 3E). The mRNA expression levels of markers for the epithelial junction, such as ZO-1, E-cadherin (E-cad), and claudin (CLDN)-1, -2, and -3 were measured to determine the effects of FMRP (Figure 3F). Compared with the DB group, the mRNA levels of ZO-1, E-cad, and CLDN-3 were significantly (*p* < 0.05) decreased in the DSS group. The magnitude of FH effects was observed in the significant (*p* < 0.05) gene expression of ZO-1, E-cad, and CLDN-1, -2. The mRNA level of ZO-1 was significantly increased and that of CLDN-2 were significantly (*p* < 0.05) decreased in the FL group compared to those in the DSS group.

### 2.3. Histopathological Changes in the Colon of the FMRP Group

Colon tissue damage was estimated by staining the colon tissue with hematoxylin and eosin (H&E) stain to analyze the pathology. Appendix A A shows that the submucosal edema (*) was more extensive in the DSS group than that in the other groups. Colonic mucosal erosions (arrow) were more common in the DSS group than those in the other groups, and submucosal swelling (*) was observed only in the DSS group (Figure 4A). FMRP (FL and FH) treatment significantly (*p* < 0.05) suppressed the average number of cell infiltrations and submucosal swelling compared to those observed in the DSS group (Appendix A).

### 2.4. FMRP (FL and FH) Regulates Enzymatic Antioxidants in the Colon Tissue

The levels of catalase (CAT), superoxide dismutase (SOD), and glutathione peroxidase (Gpx) in the DSS group were significantly (*p* < 0.05) lower than those in the DB group (Figure 4). The CAT activity and correspondingly, the reduction efficiency of H_2_O_2_ was significantly (*p* < 0.05) downregulated in the DSS group (43.33 ± 4.61 μM H_2_O_2_/g protein) compared to that in the DB group (56.00 ± 12.30 μM H_2_O_2_/g protein). However, the FL (61.92 ± 6.15 μM H_2_O_2_/g protein) and FH group (65.87 ± 8.49 μM H_2_O_2_/g protein) showed a significant (*p* < 0.05) upregulation in the activity compared to the DSS group (*p* < 0.05, Figure 4A). The SOD activity in the colon tissue of the DB (0.11 ± 0.03 U/mg protein) and FH groups (0.08 ± 0.02 U/mg protein) significantly (*p* < 0.05) increased compared to the DSS group (0.05 ± 0.01 U/mg protein) (Figure 4B). There was no significant difference in the colon tissue between the FL (0.06 ± 0.01 U/mg protein) and FH groups. A significant difference in the Gpx activity was observed between the DB (15.50 ± 8.71 mM NADPH/g protein) and DSS (5.03 ± 4.26 mM NADPH/g protein) groups. There was no significant difference in the Gpx activity in the FL (12.13 ± 4.96 mM NADPH/g protein) and FH (11.61 ± 4.33 mM NADPH/g protein) groups compared with both the DB and DSS groups (Figure 4C).

### 2.5. FMRP Altered the Cecal Microbial Diversity and Distributions at the Phylum Level

Cecal microbial diversity and composition of all groups are shown through operational taxonomic unit (OTU) levels and Chao1, Shannon, and Inverse Simpson indices (Appendix A). Microbial richness indices, such as OTUs and Chao1, were significantly (*p* < 0.05) different in the DB group compared with those of the DSS, FL, and FH groups. However, the microbial diversity index (Shannon) of the FL group was significantly (*p* < 0.05) decreased compared with that of the DB group. There was no significant difference in the inverse Simpson index between the groups. The community structure of the microbiota between all groups was analyzed using principal coordinate analysis (PCoA) based on unweighted UniFrac distance in 3D (Figure 5A). Figure 5A shows the cluster of each group (PC1 35.0%, PC2 18.1%, and PC3 12.9% plots) based on 3D PCoA for observation of the distinct separation of the DB group (red circle) vs. DSS- treated groups (DSS, FL, and FH). Administration of FMRPs slightly altered the composition of cecal microbiota compared with that observed in the group treated with DSS only. The distribution of the microbiome at the phylum level is shown in Figure 5B. Moreover, the abundance ratio of *Firmicutes* relative to *Bacteroidetes* (F/B) significantly (*p* < 0.05) increased in the DSS group compared to that in the DB group. The F/B ratio in the FL-treated group was significantly (*p* < 0.05) lower than that in the DSS group (Figure 5C).

### 2.6. FMRP Alters the Cecal Microbial Community and Distributions at the Genus Level

The microbial taxonomic community is presented with the top relative abundance of 15% of all analyzed genera (Figure 6). Among the genera belonging to the top 10 in Figure 6A, the differences in relative abundance of genera except *Prevotella* and *Mucispirillum* were listed in Figure 6B. The relative abundance in eight genera showed that there were significant or slight differences between FMRP and DSS groups. The relative abundance of *Barnesiella* in *Bacteroidetes* and *Parabacteroides* in *Bacteroidetes* increased in the FMRP group compared with that in the DB and DSS; however, the difference was not significant. The relative abundance of *Lachnoclostridium* in *Firmicutes*, *Bacteroides* in *Bacteroidetes*, and *Christensenella* in *Firmicutes* significantly (*p* < 0.05) decreased in the FMRP group compared to that in the DSS group. The genera abundance of *Lactobacillus* in *Firmicutes* also decreased in the FMRP group compared to that of the DSS group, but the difference was not significant. With respect to *Muribaculum* in *Bacteroidetes* and *Alistipes* in *Bacteroidetes*, there were no significant differences between the DSS and FMRP groups; however, relative abundance was observed in the FMRP group.

### 2.7. Analysis of Biomarker Discovery in the Gut Microbiota of the DB/DB Mice with Colitis

To identify the specific bacterial taxa as biomarkers for improving the symptoms of diabetes with colitis, the relative abundances were analyzed using linear discriminant analysis (LDA) effect size (LEfSe) based on their LDA scores between groups. A cladogram representing the differential abundance of the phylogenetic distribution in the cecum microbiota and their predominant bacteria is shown in Figure 7A,C. Members belonging to *Bacteroidetes* were highly enriched in the DB group, whereas the predominant phyla in the DSS group belonged to *Proteobacteria* (Figure 7A). Compared to the DSS and FMRP groups, members belonging to *Lachnospiraceae* were highly enriched in the DSS group (Figure 7C). Figure 7B,D show significantly different genera indicated by the LDA score higher than 2.0, the threshold between the respective groups (DB vs. DSS, and DSS vs. FMRP). Pairwise comparison between the genera of the DB and DSS groups demonstrated an increased relative abundance of *Bacteroides*, *Muricomes*, *Defluviitalea*, *Proteobacteria*, *Ruminococcus*, and *Christensenella* in the DSS group. The DB group was relatively enriched with genera such as *Muribaculum*, *Bacteroidetes*, *Alistipes*, *Prevotella*, and *Eisenbergiella* compared with the DSS group (Figure 7B). A pairwise comparison between the genera of DSS and FMRP groups showed that the relative abundance of *Harryflintia*, *Murimonas*, *Lachnoclostridium*, *Defluviitalea (s_saccharophila)*, *Muricomes*, and *Bacteroides* decreased compared with that observed in the DSS group. *Hungateiclostridium (s_straminisolvens)* and *Alistipes* were highly enriched in the FMRP group compared to those in the DSS group (Figure 7D).

### 2.8. Effects of FMRP on the Levels of Short-Chain Fatty Acids (SCFAs) in Cecum Obtained from DB/DB Mice with Colitis

In this study, SCFAs, including acetic acid (AA), butyric acid (BA), propionic acid (PA), and total SCFAs were analyzed (Appendix A). The total SCFAs were the sum of the three SCFAs. There were no significant differences in all SCFAs between the DB (AA; 142.58 ± 4.68, BA; 22.14 ± 4.78, and PA; 37.24 ± 1.41 mM/g) and DSS (AA; 183.00 ± 26.32, BA; 32.06 ± 6.28, and PA; 41.59 ± 5.87 mM/g) groups. The levels of all SCFAs in the FH (AA; 240.92 ± 29.13, BA; 49.96 ± 11.70, and PA; 57.56 ± 8.24 mM/g) group were significantly (*p* < 0.05) higher than those in the DB and DSS groups. The levels of SCFAs in the FL (AA; 198.78 ± 19.39; BA; 39.89 ± 2.10, and PA; 53.45 ± 5.01 mM/g) group significantly (*p* < 0.05) increased compared to those in the DB group; however, there was no significant difference compared with the levels observed in the DSS group. Levels of three SCFAs within the FL and FH groups showed no significant differences.

## 3. Discussion

Although the correlation between DM and IBD is being discussed, studies on the co-occurrence of DM and IBD are insufficient [8]. The gut microbiota plays an important role in diseases including IBD and DM, and recently, fecal microbiota transplantation (FMT) has been employed as a therapy, although there are still concerns regarding the safety of FMT [30]. Treatment approaches that induce changes in the microbial environment through diet without safety concerns are becoming increasingly relevant [16]. Synbiotics reflect the combination of prebiotics and probiotics, bacterial postbiotics represent metabolites, such as SCFAs and peptides, and enzyme modifications in the intestinal microbiota ecosystem [22]. A new therapeutic approach is being employed that can cure IBD using pre-, pro-, post-, and syn-biotics [31,32]. MRPs function as prebiotics and synbiotics elicit therapeutic effects in IBD [17,28].

The colitis-induced T2DM animal model (DB/DB) obtained using DSS, regarded as a useful method for representing IBD pathogenesis [29,33], was used to analyze the potential effects of FMRP on DM with colitis. The plasma insulin response, on the other hand, is a combination of two variables: insulin secretion by pancreatic β-cells and insulin metabolic clearance rate [34]. Furthermore, increased insulin concentration in serum, implying enhanced insulin resistance, is one of the major markers that distinguish metabolic disorders, such as DM and obesity [35]. The insulin concentration in the FL and FH groups returned to that of the DB group, and the IPGTT AUC level in the FL group, but not in the FH group, reduced when compared to the other groups. Although the precise mechanism for why the high dose of FMRP had a lower effect on glucose tolerance than the low dose is unknown, it suggests that the combination of insulin secretion and insulin metabolic clearance rate would change with FMRP dosage. In addition, there have been several studies that suggested insulin sensitivity was also impacted by gut microbiota [36].

Pro-inflammatory macrophage accumulation is an important feature of IBD, including ulcerative colitis [37]. In our previous experiment, RAW264.7 macrophages treated with LPS produced less NO when pre-treated with FMRP, and because it inhibited the phosphorylation of ERK and JNK in response to LPS-induced inflammation in RAW264.7 cells, fermented MRP by 4M13 showed an anti-inflammatory effect, reducing NO production and downregulating TNF- and COX-2 gene expression [26]. Nitric oxide is often released in increased amounts in IBD, which plays a deleterious role in IBD, but NO may also exert a beneficial effect against colitis according to recent research [38]. Moreover, NO generation shows protective effects in patients with DM [39]. Although pathological immune modulation has been investigated in diseases including T2DM and IBD [40,41,42], studies remain incomplete on the complex diseases. Our biochemical serum predictor results showed that certain pathogenetic processes of T2DM intensified with the induction of colitis. FH treatment was effective for the preservation of colon length, colitis evaluation, and intestinal permeability. MPO activity is commonly used as a surrogate marker of colorectal inflammation and is positively correlated with the levels of pro-inflammatory cytokines and disease severity [29]. In both FL and FH groups, the expression of IL-1β and IFN-γ, which are pro-inflammatory cytokines, and iNOS, which is preliminarily activated by pro-inflammatory cytokines, was increased as in the DB group. Pro-inflammatory cytokines in colons with colitis are often highly expressed, but this study was conducted using a combination disease model of IBD. Systemic circulating leptin deficiency in malnutrition is linked with colitis due to faulty development of cytokines, and leptin usually increases immune response by activating and proliferating Th1 cells and mediating the secretion of pro-inflammatory cytokines [43]. Therefore, the expression of partial pro-inflammatory cytokines could be downregulated in DSS-induced colitis in db/db mice relative to that in db/db mice without colitis.

ZO-1 and CLDNs are representative tight junction factors, and E-cadherin is the main adherens junction factor in the intestinal epithelia. Intestinal permeability is caused by a loss of CLDN-1 induced by E-cad deficiency and increased expression of CLDN-2 [44]. Through the evaluation of colonic histomorphometry and antioxidant enzyme activity, we observed that treatment with FMRP improved IBD in the present study. In part, inflammatory cell infiltration has led to submucosal edema and collagen fiber disruption [45]. Oxidative stress contributes to the initial stage of the progression of colorectal diseases, and antioxidant enzymes such as CAT, SOD, and Gpx play important roles in IBD. The levels of these antioxidant enzymes, as superoxide radical scavengers, are decreased in the colon of IBD patients [46].

We used 16s rRNA to see whether the regulated dysfunctions of complex diseases were caused by alterations in the gut microbial composition in IBD patients with diabetes. The FMRP treatment significantly (*p* < 0.05) recovered the *Firmicute*/*Bacteroidetes* ratio in the DB group. The relative abundance of microbial composition in metabolic disorder and IBD was compatible with previous research [47,48]. *Bacteroidetes* is mainly associated with healthy gut microbiota composition [49]. *Bacteroides* spp. have been found in approximately 30% of *Bacteroidetes* phyla, and the population of *Bacteroides* spp. has been demonstrated as a diagnostic criterion for IBD [50]. *Proteobacteria*, which comprise a wide variety of pathogens, were the predominant bacterial phyla in the gut microbiota of CD patients [13]. In the present study, the relative genus taxa such as *Harryflintia*, *Murimonas*, *Lachnoclostridium*, *Defluviitalea*, *Muricomes*, and *Bacteroides* were more enriched in the DSS group than those in the FMRP group. These results are consistent with those of the gut microbial analysis in metabolic disorders and IBD performed by many researchers. In a recent study, the increased proportion of *Defluviitalea saccharophila* in obese and diabetic mice was confirmed [51]. Family *Lachnospiraceae* of *Lachnoclostridium*, which was the predominant taxa of the DSS group in our study, was positively correlated with T2DM [52]. Moreover, the upregulation of the microbial abundance of *Lachnoclostridium* was observed in colorectal cancer patients [53]. Both *Murimonas intestini* and *Muricomes intestini* belong to *Lachnospiraceae*, and *Muricomes intestini* is associated with liver weight and liver LDL-cholesterol [54,55]. These results indicate that treatment with FMRP decreased the intestinal abundance of the bacteria associated with T2DM and IBD. In addition, the *Alistipes* genus is known to be a colitis-attenuating genus and a producer of propionate and butyrate in IBD [56,57]. *Alistipes finegoldii* can efficiently construct membrane lipids using medium-chain fatty acids in the gut epithelium [58]. *Hungateiclostridium straminisolvens*, previously classified as CSK1, which is a cellulolytic bacterium, has not been studied sufficiently; however, it produces acetate and isopropanol [59]. More research into the beneficial effects of *Hungateiclostridium straminisolvens* on colitis, however, is required. Likewise, the amounts of SCFA-producing bacteria, such as *Alistipes* and *Hungateiclostridium*, were significantly increased in the cecal samples after the administration of FMRP. Metabolites play a key role in the host immune system and gut bacterial dysbiosis. SCFAs play an important role in metabolic functions, intestinal health, and microbial composition. Furthermore, SCFAs improve insulin sensitivity and reduce body weight [60]. As a result of these considerations, we propose that FMRP administration can modulate the gut microbiota and improve dysfunctions in complex diseases (Figure 8).

## 4. Materials and Methods

### 4.1. Chemicals and Materials

DSS (MW: 36–50 kDa) was purchased from MP Biomedical, LLC (Illkirch-Graffenstaden, France). Hydrogen peroxide (H_2_O_2_) (30%) was purchased from Junsei Chemical Co. Ltd. CAT, Gpx, SOD, *o*-dianisidine, and hexadecyltrimethylammonium bromide (HTAB) were purchased from Sigma Aldrich (St. Louis, MO, USA). FITC-dextran was purchased from Sigma-Aldrich (St. Louis, MO, USA). MTBSTFA was purchased from Sigma-Aldrich (St. Louis, MO, USA). The standards of acetic, butyric, and propionic acids were purchased from Sigma Aldrich (St. Louis, MO, USA). Acetic acid-d4 was purchased from Sigma-Aldrich (St. Louis, MO, USA).

### 4.2. Sample Preparation

MRP was conjugated with a whey protein isolate and galactose. To prepare FMRP, the liquid phase of MRP was fermented by *L*.4M13 for 48 h at 37 °C. *L*.4M13 was provided by the Seoul Dairy Cooperative (4M13, R&D Center, Seoul Dairy Cooperative, Kyunggi, Republic of Korea). FMRP was stored at −80 °C after lyophilization.

### 4.3. Experimental Design and Treatment

Six-week-old male C57BLKS/J (−/−) mice were purchased from Central Lab Animal Inc. (Seoul, South Korea). All C57BLKS/J (−/−) (DB/DB) mice weighing 25–34 g were divided in the experimental groups DB, DSS, FL, and FH. Nine mice were randomly allocated in each group and maintained in individual cages. After one week of acclimation, all groups were treated orally for 14 d. The DB and DSS groups were orally administered with PBS. The FL and FH groups were orally treated with FMRP low (FL; 750 mg/kg b.w.) and high (FH; 1500 mg/kg b.w.) concentrations, respectively. Mice were fed 2% DSS water ad libitum for 5 d prior to euthanization in the DSS, FL, and FH groups for induced colitis. The experiment was conducted under standard conditions at the KU-GEAR center (12 h light/12 h dark cycle, temperature 25 ± 2 °C, and humidity 70–75%). The experimental design was approved by the International Animal Care and Use Committee (IACUC No. KUIACUC-2018-24) of Korea University, Seoul, Korea. All biological samples (serum, colon tissue, and cecum) were collected after euthanization and stored at −80 °C until the end of the study.

### 4.4. Assessment of Permeability and Disease Activity Index in Colitis

FITC-dextran was used to measure the colon permeability. The permeability assay was modified based on a recent study [62]. The mice were orally administered FITC-dextran with a dose of 200 mg/kg body weight 4 h before euthanasia. After euthanizing, the serum from mice was collected and maintained at −80 °C before further experiments. Fluorescence of FITC-dextran was measured in serum at an excitation wavelength of 480 nm and an emission wavelength of 560 nm using a VICTOR3™ spectrofluorometer (PerkinElmer Inc., Waltham, MA, USA). DAI in colitis was measured on 5 consecutive days with 2% DSS. The DAI scores were estimated with the following parameters: degree of loose stool (0–1: normal, 2–3: loose stool, 4: diarrhea), and degree of bleeding (0, normal; 1, ±; 2, +; 3, ++; 4, gross). The DAI score measurement and calculation were performed using the methods described by Hong et al. [17].

### 4.5. Determination of the Levels of Insulin and TNF-α and NO Production in the Serum

Serum insulin concentration was analyzed using a commercial ELISA kit (RayBiotech, Inc., Norcross, GA, USA). The level of TNF-α in mouse serum was measured using a commercial ELISA kit (Mouse TNF-alpha, pink-one, Komabiotech, Seoul, Korea). Insulin and TNF-α levels were measured according to the manufacturers’ protocols. The amount of NO in mouse serum was measured using the Griess reaction [26].

### 4.6. Determination of MPO Activity

The amount of MPO generated in damaged colon tissue was measured using the modified protocol of Kim et al. [29]. The colon tissue was homogenized in a buffer containing HTAB and o-dianisidine dihydrochloride. MPO activity was measured at 450 nm and 30 s intervals [33].

### 4.7. Histological Evaluation of Colitis

Hematoxylin and eosin (H&E) staining was used to evaluate the pathological damage of the colon tissue obtained from mice with colitis. The colon tissues were immobilized in 4% formaldehyde. The tissues were dehydrated and embedded in paraffin. The paraffin blocks were sliced into 5 µm-thick sections and dried with xylene to remove the paraffin on a slide. Tissue sections were stained with H&E. Morphological changes were visualized under a light microscope. The cellular infiltration of the mucosa and swelling of the submucosal mucosa were observed.

### 4.8. Determination of Gpx, SOD, and CAT in Colon Tissues

For the Gpx, SOD, and CAT assays, the colon tissue was homogenized with cold microsome lysis buffer (Tris-HCl, KCl, and EDTA with 0.1 M potassium phosphate buffer) using an ultrasonicator. After centrifugation at 12,000× *g* for 30 min at 4 °C, each supernatant was transferred to a new tube and stored at −80 °C until further analysis. The activities of Gpx, SOD, and CAT were determined using the method published by Hong et al. with minor modifications [17].

### 4.9. Quantitative Reverse Transcriptase PCR Analysis

To analyze the expression levels of cytokines and genes that are associated with the epithelial junction in the colon of mice belonging to all groups, total RNA was isolated from the colon tissue using RNAiso plus reagent (Takara Japan), and cDNA was synthesized from 2 μg of total RNA using the LeGene Premium Express 1st strand cDNA synthesis system (LeGene, San Diego, CA, USA). Quantitative reverse transcriptase PCR (qRT-PCR) was performed with a Bio-Rad iQ5 thermal cycler according to the manufacturer’s instructions using the real-time SYBR Green method (Bio-Rad, Hercules, CA, USA).

### 4.10. Colonic Microbiota Analysis

Cecal DNA was extracted for the analysis of microbial composition. The Hypervariable V3-V4 region of 16s rRNA amplicons was generated using the MiSeq (Illumina, San Diego, CA, USA) at Macrogen (Seoul, Korea) following the manufacturer’s instructions. The alpha-diversity analysis was performed using quantitative insights with the microbial ecology (QIIME) software, and operational taxonomic units (OTUs) were defined at ≥97% sequence homology. The beta diversity distance matrix based on the unweighted UniFrac metric was calculated using PCoA followed by performing permutational multivariate analysis of variance (MANOVA). Taxonomic composition was analyzed using QIIME-UCLUST based on the Ribosomal Database Project.

### 4.11. Cecal SCFA Extraction and Derivatization

Cecal samples were collected after euthanization and stored at −80 °C. SCFA concentrations of cecal samples were measured using gas chromatography–mass spectrometry (GC/MS). Cecal contents (10 mg) were homogenized with 250 μL of the extract solution (200 μL ether and 50 μL HCl) and 100 μL of the internal standard (acetic-d4 acid). After homogenization and shaking for 20 min, the homogenates were centrifuged at 1000× *g* for 10 min. The supernatant (80 μL), which was transferred into a glass insert vial, was mixed with 16 μL of MTBSTFA. The screw vial was sealed with an insert vial, incubated at 60 °C for 20 min in a dry oven, and then incubated at 25 °C for 48 h for derivatization [63].

### 4.12. GC/MS Analysis

The derivatized samples were analyzed using a 6890N network GC system (Agilent Technologies) equipped with an HP-5MS UI column (0.25 mm × 30 m× 0.25 μm, Agilent Technologies, Santa Clara, CA, USA) and 5975 network mass selective detector (Agilent Technologies). Helium (99.999%) was used as a carrier gas at a constant flow rate of 1.2 mL/min. The identification of compounds was performed by injecting pure standards with known retention time. The GC conditions were set as follows: injection volume, 1.0 μL; head pressure, 97 kPa; split 20:1; inlet and transfer line temperatures, 250 and 260 °C, respectively. The initial oven temperature was 60 °C for 3 min and increased by 20 °C/min from 120 to 300 °C after 5 °C/min to 120 °C. The data were quantified in selected ion monitoring (SIM) mode using the target ion. The target ions (*m*/*z*) of acetic-d4, acetic, propionic, and butyric acids were 120, 117, 131, and 145, respectively [63,64].

### 4.13. Statistical Analysis

All results except those obtained after microbiome analysis are represented as mean ± standard deviation (SD); all experiments (except microbiome analysis that was performed with triplicate samples) were performed with five or six replicates. Differences between groups were analyzed with one-way ANOVA using Tukey’s or Duncan’s studentized range test.

## 5. Conclusions

Our results show that FMRP administration could improve insulin resistance by increasing the levels of SCFAs and reducing the amount of microbiota associated with DM. The model with mice used in this study may not fully reflect the human chronic IBD. Additionally, the direct signaling pathway between gut microbiota and dysfunction factors in the co-occurrence of DM and IBD, however, remains unknown. In UC patients, probiotics have been shown to help induce and sustain remission [61]. Diet may influence the composition of intestinal flora and microbial metabolites [61], and MRPs can be used as prebiotics and synbiotics to treat IBD [17,28]. Collectively, we suggest that FMRP treatment can be used as a pharmaceutical supplement to modify the microbial composition of the intestines in a healthy way and produce metabolites that improve metabolic functions.

## Figures and Tables

**Figure 1 pharmaceuticals-14-00299-f001:**
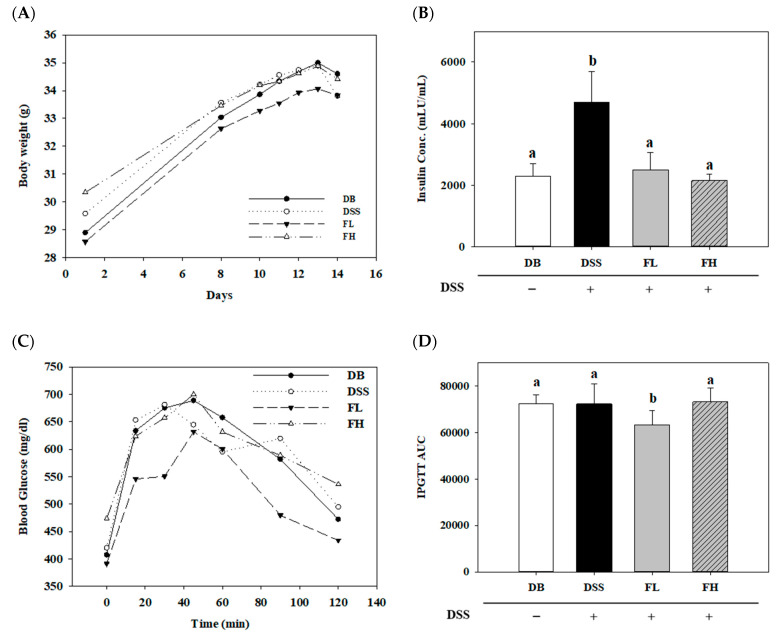
Effect of fermented Maillard reaction product (MRP) formed by *Lactobacillus gasseri* 4M13 (*L*.4M13) (FMRP) on the changes in the body weight (**A**), serum insulin concentration (**B**), blood glucose level (**C**), and intraperitoneal glucose tolerance (IPGTT) area under the curve (AUC) (**D**) in the serum. The FL and FH groups were orally administered with low (FL; 750 mg/kg b.w.) and high (FH; 1500 mg/kg b.w.) FMRP concentrations, respectively. Mice were fed 2% dextran sulfate sodium (DSS) in water ad libitum for 5 d prior to euthanization in the DSS, FL, and FH groups for induced colitis. DSS +/− indicates whether 2% DSS is treated or not. Different letters (a–c) denote a significant difference (*p* < 0.05) calculated via one-way ANOVA with Tukey’s multiple comparison test. Data are expressed as means ± SD. Six mice were included per group.

**Figure 2 pharmaceuticals-14-00299-f002:**
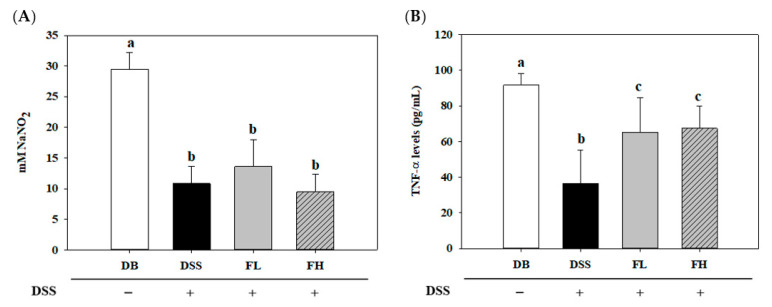
Effect of FMRP on nitric oxide (NO) (**A**) and TNF-α levels (**B**) in the serum. The DSS, FL, and FH groups were voluntarily administered 2% DSS in water ad libitum for 5 d. DSS +/− indicates whether 2% DSS is treated or not. Different letters (a–c) denote a significant difference (*p* < 0.05) calculated using one-way ANOVA with Tukey’s multiple comparison test. Data are expressed as means ± SD. Six mice were included per group.

**Figure 3 pharmaceuticals-14-00299-f003:**
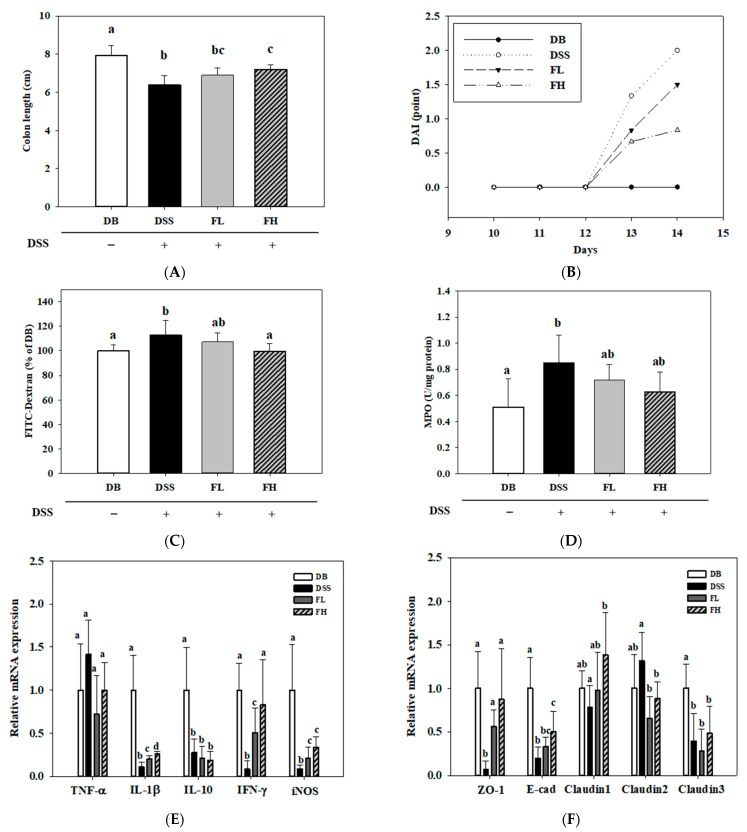
Effects of FMRP on the colon length (**A**), disease activity index (DAI) (**B**), intestinal permeability (**C**), myeloperoxidase (MPO) (**D**), mRNA expression of inflammatory factors (**E**), and epithelial junction markers (**F**) in the colon tissue. DSS +/− indicates whether 2% DSS is treated or not. Different letters (a–c) denote a significant difference at *p* < 0.05 calculated using one-way ANOVA with Tukey’s multiple comparison test. Data are expressed as means ± SD. Six mice were included per group.

**Figure 4 pharmaceuticals-14-00299-f004:**
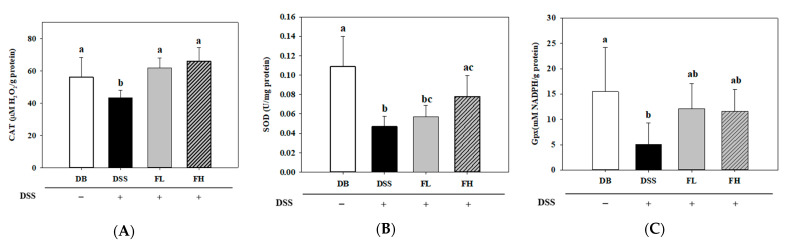
Effects of FMRP on the antioxidant enzyme activity in the colon tissue. (**A**) Catalase (CAT) activity, (**B**) Superoxide dismutase (SOD) activity, and (**C**) Glutathione peroxidase (Gpx) activity. DSS +/− indicates whether 2% DSS is treated or not. Different letters (a–c) denote a significant difference at *p* < 0.05 calculated using a one-way ANOVA with Duncan’s multiple comparison test. Data are expressed as means ± SD. Five mice were included per group.

**Figure 5 pharmaceuticals-14-00299-f005:**
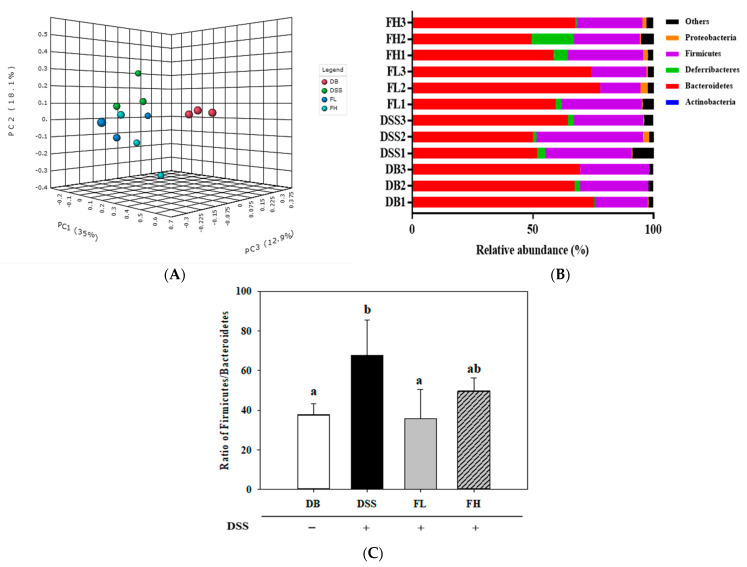
FMRP modulates the gut microbiota composition in colitis induced in the DB/DB mice. (**A**) The 3D principal coordinate analysis (PCoA) plots unweighted UniFrac based on operational taxonomic unit (out) level followed with permutational multivariate analysis of variance (PERMANOVA). (**B**) Relative abundance plot of the bacterial phylum level. (**C**) The ratio of *Firmicutes* to *Bacteroidetes*. DSS +/− indicates whether 2% DSS is treated or not. Different letters (a–b) denote a significant difference at *p* < 0.05 calculated using one-way ANOVA with Tukey’s multiple comparison test. Data of three cages per group are expressed as means ± SD. Three cages with 9 mice were included per group.

**Figure 6 pharmaceuticals-14-00299-f006:**
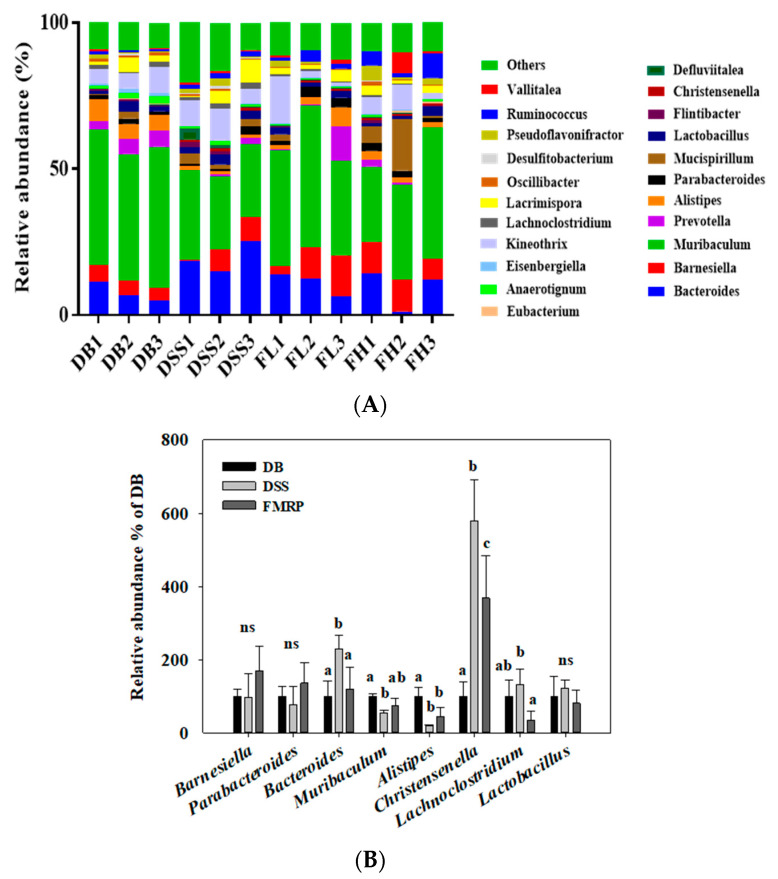
FMRP modulates the genus-level operational taxonomic units in DB/DB mice with colitis. (**A**) Relative abundance (%) plot of the DB group at the bacterial genus level. (**B**) Changes of specific genera; *Barnesiella* in *Bacteroidetes,*
*ParabacTable 6.* group included the FL (*n* = 3) and FH (*n* = 3) groups. The different letters (a–c) denote a significant difference at *p* < 0.05 calculated using one-way ANOVA with Duncan’s multiple comparison test. ‘ns’ refers to no significance.

**Figure 7 pharmaceuticals-14-00299-f007:**
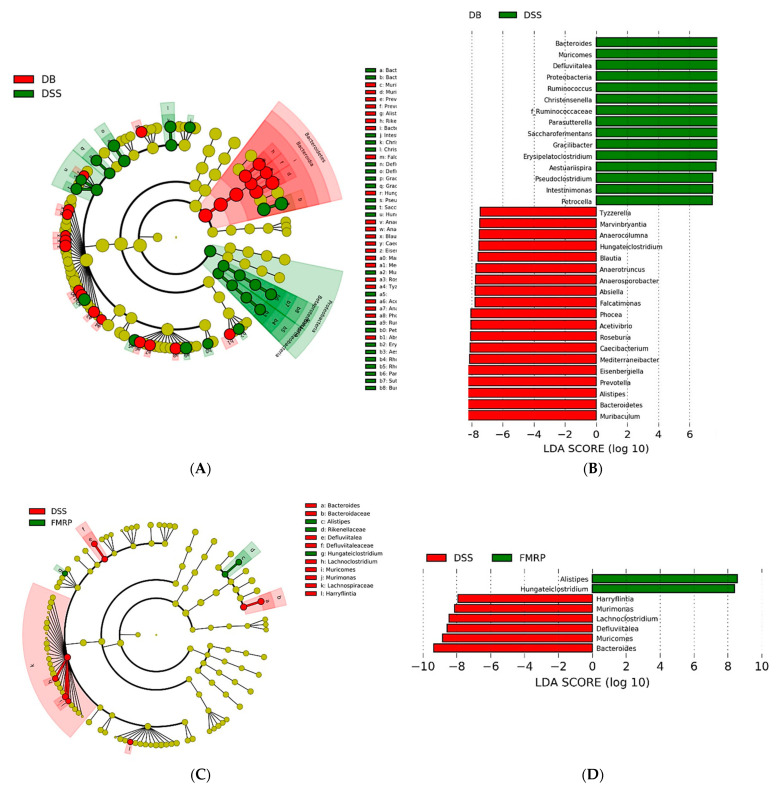
Linear discriminant analysis (LDA) effect size (LEfSe) was performed to examine changes in the gut microbiota following type 2 diabetes mellitus (DB), DB with colitis (DSS), and DSS with FMRP treatment (FL and FH). LEfSe cladogram shows the most differentially abundant taxa between the DB and DSS groups (**A**) and the DSS and FMRP groups (**C**). The taxonomic cladogram was obtained from the LEfSe analysis of 16S rRNA sequences. Bars represent the effect size for each taxon between the DB and DSS groups (**B**) and the DSS and FMRP groups (**D**). The length of the bar represents the log10-transformed LDA score. The threshold of the logarithmic LDA score for discriminative features was set to 2.0. The taxon of bacteria at the genus level with statistically significant change (*p* < 0.05) in the relative abundance is shown alongside the horizontal lines. Taxa enriched for FMRP are shown in green, and DSS-enriched taxa are shown in red.

**Figure 8 pharmaceuticals-14-00299-f008:**
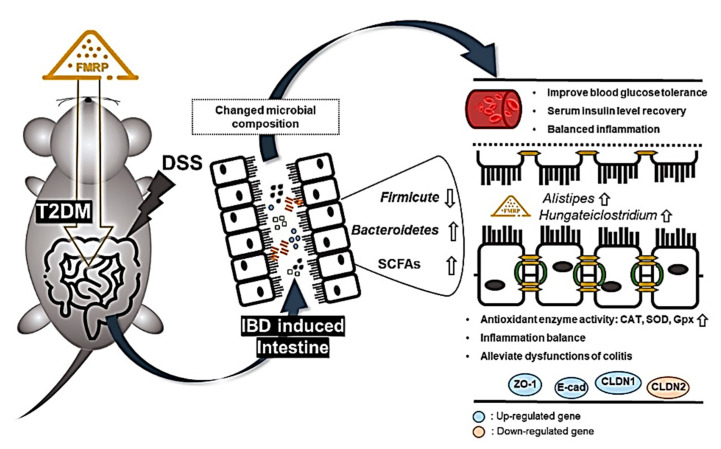
Scheme depicting the proposed mechanisms for improving type 2 diabetes (T2DM) dysfunctions associated with inflammatory bowel disease (IBD) through microbiota intestinal changes in FMRP administration. The ↑ denotes up-regulation and the ↓ means down-regulation.

## Data Availability

The data presented in this study are available on request from the corresponding author.

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
