# Peer review of "Fermented Maillard Reaction Products by Lactobacillus gasseri 4M13 Alters the Intestinal Microbiota and Improves Dysfunction in Type 2 Diabetic Mice with Colitis"

_pharmaceuticals, 2021, doi:10.3390/ph14040299_

Round 1

Reviewer 1 Report

In this study, Yu-Jin Jeong, et al. investigated the effects of the fermented glycated conjugate of 65 whey protein and galactose with Lactobacillus gasseri 4M13 (FMRP) in rats with Type 2 Diabetes mellitus with colitis. They conclude that FMRP administration could improve insulin resistance by increasing the levels of SCFAs and reducing the amount of microbiota associated with DM.

Strengths of this study:

Well designed study.

Research question is valid.

Adequate literature review was performed.

The author's major findings were clearly presented. They adequately address the stated research objectives.

The research results validate the author's conclusions.

I am overall pleased with the study design and results except for the following issues.

  • Authors may want to mention limitations of this study and include expansion of abbreviation PBS.

Author Response

Reviewer1:

In this study, Yu-Jin Jeong, et al. investigated the effects of the fermented glycated conjugate of 65 whey protein and galactose with Lactobacillus gasseri 4M13 (FMRP) in rats with Type 2 Diabetes mellitus with colitis. They conclude that FMRP administration could improve insulin resistance by increasing the levels of SCFAs and reducing the amount of microbiota associated with DM.

Strengths of this study:  Well designed study.

Research question is valid.

Adequate literature review was performed.

The author's major findings were clearly presented. They adequately address the stated research objectives.

The research results validate the author's conclusions.

 I am overall pleased with the study design and results except for the following issues.

Comments: 

Authors may want to mention limitations of this study and include expansion of abbreviation PBS.

Þ Answer:

First of all, we’d like to express our gratitude to the reviewer for the careful and critical reading of our manuscript. With highlighted in green color text, we made the correction in response.

We mentioned limitations of this study on page (P) 12, lines (L) 368-370 and added the full name of PBS to P3, L80 as follows:

(P12, L368-370)

“The model with mice used in this study may not fully reflect the human chronic IBD. Also, the direct signaling pathway between gut microbiota and dysfunction factors in the co-occurrence of DM and IBD, however, remains unknown.”

(P3, L80)

“All groups were treated with phosphate buffered saline (PBS) or FMRP orally for 14 d, and the changes in body weight were examined daily after the ad libitum administration of DSS water (Figure 1A).”

Reviewer 2 Report

In this manuscript, the authors observed that fermented products by 4M13 could prevent T2DM with IBD. However, the authors need to provide additional data to support their findings.

  1. The authors published a 4M13 related paper that mentioned about that fermented product (F-GWA) from 4M13 had an anti-inflammatory effect on lipopolysaccharide (LPS)-stimulated RAW264.7 cells (J Dairy Sci. 2019 Sep;102(9):7707-7716). However, in this manuscript, the authors did not discuss about the correlation between macrophage and colitis.
  2. Previous study showed that Lactobacillus gasseri SF1183 affects intestinal epithelial cell survival and growth (PLoS One. 2013 Jul 23;8(7):e69102). Why the authors did not direct to use 4M13? Maybe the prevention effect will be better than use fermented products.
  3. It is very important to show the fermented products by 4M13 could prevent wild-type mice with IBD or not. Could the authors show relation data?
  4. In figure 1, please discuss about why the insulin concentration decreased in FL and FH groups but the IPGTT did not change in FH groups.
  5. There was no direct evidence that Alistipes and Hungateiclostridium increased could improve the colitis. The authors need to treat the mice with Alistipes and Hungateiclostridium directly to verity these results or show the previous study results.
  6. Please use the picture to show the mechanism that fermented products by 4M13 involved to improve the colitis in T2DM mice.

Author Response

Reviewer2:

In this manuscript, the authors observed that fermented products by 4M13 could prevent T2DM with IBD. However, the authors need to provide additional data to support their findings.

Comments:

1) The authors published a 4M13 related paper that mentioned about that fermented product (F-GWA) from 4M13 had an anti-inflammatory effect on lipopolysaccharide (LPS)-stimulated RAW264.7 cells (J Dairy Sci. 2019 Sep;102(9):7707-7716). However, in this manuscript, the authors did not discuss about the correlation between macrophage and colitis.

Þ Answer:

First of all, we’d like to express our gratitude to the reviewer for the careful and critical reading of our manuscript. With highlighted in green color text, we made the correction in response.

We added content about the correlation between macrophage and colitis in page (P) 11, line (L) 298-303 as follows:

“Pro-inflammatory macrophage accumulation is an important feature of IBD, including ulcerative colitis [37]. In our previous experiment, RAW264.7 macrophages treated with LPS produced less NO when pre-treated with FMRP, and because it inhibited the phosphorylation of ERK and JNK in response to LPS-induced inflammation in RAW264.7 cells, FMRP showed an anti-inflammatory effect, reducing NO production and downregulating TNF- and COX-2 gene expression [26].”

2) Previous study showed that Lactobacillus gasseri SF1183 affects intestinal epithelial cell survival and growth (PLoS One. 2013 Jul 23;8(7):e69102). Why the authors did not direct to use 4M13? Maybe the prevention effect will be better than use fermented products.

Þ Answer:

Thank you for your comment. There were several previous studies that determined the more beneficial effects of fermented products or symbiotic on diseases than probiotics. We added the contents about the effects of fermented products on P2, L61-68 as follows:

Lactobacillus gasseri 4M13 (L.4M13) strains, which were isolated from infant feces and were used in present study as probiotics, have functional properties such as inhibitory effects on α-glucosidase activity and nitric oxide (NO) production, anti-oxidants and immune enhanced effects [24]. Microbial fermentation of MRPs with probiotics has more powerful effects such as anti-inflammation, antioxidant effects anti-IBD, and anti-microbial activity [25-27]. The conjugates demonstrate high potential as prebiotics, and their fermented products obtained by the action of probiotics are more effective [28].”

3) It is very important to show the fermented products by 4M13 could prevent wild-type mice with IBD or not. Could the authors show relation data?

Þ Answer:

We appreciate the reviewer’s comment again. We’ve already performed experiment that the fermented products by 4M13 prevent wild-type mice with IBD. The manuscript (title: Homeostasis effects of fermented Maillard reaction products in dextran sulfate sodium-induced colitis mice) was recently submitted to ‘Journal of the Science of Food and Agriculture’. The manuscript has been reviewed now.

In the text, we added the concerned statement in P4 L113-114 as follows:

“In the wild-type mice with IBD, the administration of FMRP alleviated colonic damage in DSS-induced colitis (submitted for publication).”

Here we provide some results of the manuscript.

The all mice were wild type, and the groups were represented that ‘CON’: treated PBS, ‘DSS’: 2% DSS for 7 days, ‘FG’: fermented galactose with 4M13, and ‘FWG’: fermented MRP conjugated with galactose and whey protein with 4M13 (equal to FMRP in this study).

Figure 1.

Figure 6.

4) In figure 1, please discuss about why the insulin concentration decreased in FL and FH groups but the IPGTT did not change in FH groups.

Þ Answer:

We discussed about why the insulin concentration decreased in FL and FH but the IPGTT did not change in FH in P11, L287-297 as follows:

The plasma insulin response, on the other hand, is a combination of two variables: insulin secretion by pancreatic β-cells and insulin metabolic clearance rate [34]. Furthermore, increased insulin concentration in serum, implying enhanced insulin resistance, is one of the major markers that distinguish metabolic disorders, such as DM and obesity [35]. The insulin concentration in the FL and FH groups returned to that of the DB group, and the IPGTT AUC level in the FL group, but not in the FH group, reduced when compared to the other groups. Although the precise mechanism for why the high dose of FMRP had a lower effect on glucose tolerance than the low dose is unknown, it suggests that the combination of insulin secretion and insulin metabolic clearance rate would change with FMRP dosage. In addition, there were several studies that insulin sensitivity was also impacted by gut microbiota [36].

5) There was no direct evidence that Alistipes and Hungateiclostridium increased could improve the colitis. The authors need to treat the mice with Alistipes and Hungateiclostridium directly to verity these results or show the previous study results.

Þ Answer:

We added the sentence about the potential effects of Alistipes genus on colitis-attenuating in IBD in P12, L352-353. But the investigation about Hungateiclostridium on colitis was insufficient. So, we suggested that the further studies are required in P12, L354-355, L359-360 as follows:

(P12, L354-355)

“In addition, the Alistipes genus is known to be a colitis-attenuating genus and a producer of propionate and butyrate in IBD [56,57].”

(P12, L359-360)

“More research into the beneficial effects of Hungateiclostridium straminisolvens on colitis, however, is required.”

6) Please use the picture to show the mechanism that fermented products by 4M13 involved to improve the colitis in T2DM mice.

Þ Answer:

Please notice that we supplemented the graphical abstract to show the beneficial effects of fermented products by 4M13 on not only changing microbiota but also dysfunction factors of both colitis and T2DM.

Reviewer 3 Report

The manuscript is interesting. Still, I would recommend authors to use abbreviations in the abstract to a lesser extent. 

Also, the Introduction should include a discussion regarding the controversial effects of MRP, since there is important literature evidence suggesting the detrimental effects of these compounds/their role in disease progression. 

Author Response

Reviewer3:

Comments:

1) The manuscript is interesting. Still, I would recommend authors to use abbreviations in the abstract to a lesser extent. 

Þ Answer:

First of all, we’d like to express our gratitude to the reviewer for the careful and critical reading of our manuscript. With highlighted in green color text, we made the correction in response.

We re-wrote for reducing abbreviations in abstract on page (P) 1, lines (L) 11-25 as follows:

Abstract: Inflammatory bowel disease is a chronic relapsing disease. Multiple factors can cause IBD, including diet, imbalance of the immune system, and impaired intestinal barrier function. Type 2 diabetes mellitus is a complex and chronic metabolic disease caused by a combination of insulin resistance and an ineffective insulin secretory response. The co-occurrence of these two diseases, demonstrating interrelated effects within the gut microbiota, has been frequently reported. This study evaluated the effects of a fermented glycated conjugate of whey protein and galactose with Lactobacillus gasseri 4M13 (FMRP) to prevent type 2 diabetes mellitus with inflammatory bowel disease. C57BLKS/J- db/db mice were orally administered FMRP for 14 consecutive days and 2% dextran sulfate sodium (DSS) in water ad libitum for 5 days to induce colitis. FMRP-fed mice showed improved insulin secretion and symptoms of colitis. Compared to the DSS group, the FMRP group showed a decreased abundance of six bacterial genera and increased abundance of Alistipes and Hungateiclostridium. In cecal contents, the levels of short-chain fatty acids increased in the FMRP group compared to those in the DSS group. Continuous administration of FMRP thus may improve the homeostasis of not only insulin secretion and inflammation, but also the intestinal environment in inflammatory bowel disease and type 2 diabetes mellitus.”

2) Also, the Introduction should include a discussion regarding the controversial effects of MRP, since there is important literature evidence suggesting the detrimental effects of these compounds/their role in disease progression. 

Þ Answer:

We added the concerned statements in P2, L56-60 as follows:

“However, it is also worth noting that the Maillard reaction can result in the formation of advanced glycation end products (AGEs), such as methylglyoxal-derived AGEs, which have been associated to oxidative stress and diabetes [21]. Thus, it is necessary to minimize quantity of AGEs. ”

Reviewer 4 Report

In this study, authors investigated multiple effects of Fermented Maillard reaction products on intestines of diabetic mice with DSS-induced colitis.

This experimental study provides new and interesting information in the area of IBD and diabetes. Gap in the literature is well described, experiments are explained in detail and overall work was executed well.

I have a few comments for the improvement of some of the manuscript aspects:

Result section 2.1. – provide full name of the PBS abbreviation and DB mice (for additional explanation for the readers)

There is quite a lot of large figures and presented results. I suggest putting some of them into supplemental materials (i.e. Figure 4, 8 or 9), with main results still explained in the manuscript text.

I find Discussion section to be long and somewhat difficult to read. I suggest forming paragraphs, and re-arranging the writing in the same order as results were presented. Furthermore, limitations of the study should be addressed.

Lastly, as DSS-colitis is acute colitis, more thoughts and implications on potential effects of FMRP regarding chronic IBD patients’ disease progression could be addressed in Discussion

Author Response

Reviewer4:

In this study, authors investigated multiple effects of Fermented Maillard reaction products on intestines of diabetic mice with DSS-induced colitis.

This experimental study provides new and interesting information in the area of IBD and diabetes. Gap in the literature is well described, experiments are explained in detail and overall work was executed well.

I have a few comments for the improvement of some of the manuscript aspects:

Comments:

1) Result section 2.1. – provide full name of the PBS abbreviation and DB mice (for additional explanation for the readers)

Þ Answer:

First of all, we’d like to express our gratitude to the reviewer for the careful and critical reading of our manuscript. With highlighted in green color text, we made the correction in response.

We added the full name of PBS and DB mice information in page (P) 3, lines (L) 80 and P3, L83 as follows:

“All groups were treated with phosphate buffered saline (PBS) or FMRP orally for 14 d, and the changes in body weight were examined daily after the ad libitum administration of DSS water (Figure 1A). To demonstrate the changes in blood glucose and insulin concentrations in C57BLKS/J- db/db mice (DB/DB) with DSS-induced colitis, the intraperitoneal glucose tolerance test (IPGTT) was performed, and the insulin levels were measured using an ELISA kit (Figure 1B-D).”

2) There is quite a lot of large figures and presented results. I suggest putting some of them into supplemental materials (i.e. Figure 4, 8 or 9), with main results still explained in the manuscript text.

Þ Answer:

Please note that we moved the Figures 4, 8, and 9 into Figures S1, S2, and S3, respectively as the reviewer 4 suggested. And figure numbers were changed in main text.

Figure S1.

(A)

(C)

   (B)

Figure S1. Histological changes in the colon following FMRP treatment. Hematoxylin and eosin (H&E)-stained photomicrographs showing the colon (×200 and ×400 Bar = 100 mm). Morphological changes were visualized using a light microscope after H&E staining (→: cell infiltration, *: submucosal swelling). Different letters (a–b) denote a significant difference at p < 0.05 calculated using one-way ANOVA with Tukey’s multiple comparison test. Data are expressed as means ± SD. Five mice were included per group.

Figure S2.

(A)

(B)

(C)

(D)

Figure S2. Microbial community analysis of cecal samples in colitis induced DB/DB mice. Alpha diversity indexes are composite indexes reflecting abundance and consistency. (A) Operational taxonomic unit (OTU) levels, (B) Chao1, (C) Shannon’s and (D) Inverse Simpson indices. The different letters (a–b) denote a significant difference at p < 0.05 using a one-way ANOVA with Tukey’s multiple comparison test. Data are expressed as means ± SD. Three cages (3 mice/cage) were included per group.

Figure S3.

(A)

(B)

(C)

(D)

Figure S3. Levels of short-chain fatty acids (SCFAs) in the cecal contents. (A) Total SCFAs, (B) acetic acid, (C) butyric acid, and (D) propionic acid were analyzed using GC-MS. Different letters (a–c) denote a significant difference at p < 0.05 calculated using one-way ANOVA with Tukey’s multiple comparison test. Data are expressed as means ± SD. Five mice were included per group.

3) I find Discussion section to be long and somewhat difficult to read. I suggest forming paragraphs, and re-arranging the writing in the same order as results were presented. Furthermore, limitations of the study should be addressed.

Þ Answer:

We deleted the overlapping sentences with results to shortening discussion section and re-arranged the discussion section in the same order as results were presented in P10-12, L273-376. And we added the limitations of the study on P12, L368-370 as follows:

“The model with mice used in this study may not fully reflect the human chronic IBD. Also, the direct signaling pathway between gut microbiota and dysfunction factors in the co-occurrence of DM and IBD, however, remains unknown.”

4) Lastly, as DSS-colitis is acute colitis, more thoughts and implications on potential effects of FMRP regarding chronic IBD patients’ disease progression could be addressed in Discussion

Answer: We added the sentence containing meaning that implications on potential effects of FMRP regarding chronic IBD patients’ disease progression in P12, L370-373 as follows:

“In UC patients, probiotics have been shown to help induce and sustain remission [61]. Diet may influence the composition of intestinal flora and microbial metabolites [61], and MRPs can be used as prebiotics and synbiotics to treat IBD [17,28]. Collectively, we suggest that FMRP treatment can be used as a pharmaceutical supplement to modify the microbial composition of the intestines in a healthy way and produce metabolites that improve metabolic functions.”

Round 2

Reviewer 2 Report

The authors have modified the concerns that I mentioned. However, the picture that show the mechanism in this study still not showed in the manuscript. Please add this part.

Author Response

Reviewer(s’) comments

Reviewer 2

Comments and Suggestions for Authors

The authors have modified the concerns that I mentioned. However, the picture that show the mechanism in this study still not showed in the manuscript. Please add this part.

Þ Answer:

First of all, we would like to thank the reviewer for the attentive and critical reading to improve our manuscript. With highlighted in green color text, we made the correction in response.

We added the Figure 8 on page 12, lines 365-369as follows:

“As a result of these considerations, we propose that FMRP administration can modulate the gut microbiota and improve dysfunctions in complex diseases (Figure 8). ”

Figure 8. Scheme depicting the proposed mechanisms for improving type 2 diabetes (T2DM) dysfunctions associated with inflammatory bowel disease (IBD) through microbiota intestinal changes in FMRP administration.
